# The Characteristic Microstructures and Properties of Steel-Based Alloy via Additive Manufacturing

**DOI:** 10.3390/ma16072696

**Published:** 2023-03-28

**Authors:** Chunlei Shang, Honghui Wu, Guangfei Pan, Jiaqi Zhu, Shuize Wang, Guilin Wu, Junheng Gao, Zhiyuan Liu, Ruidi Li, Xinping Mao

**Affiliations:** 1Beijing Advanced Innovation Center for Materials Genome Engineering, Innovation Research Institute for Carbon Neutrality, University of Science and Technology Beijing, Beijing 100083, China; 2Department of Environmental and Municipal Engineering, North China University of Water Resources and Electric Power, Zhengzhou 450046, China; 3Additive Manufacturing Institute, College of Mechatronics and Control Engineering, Shenzhen University, Shenzhen 518060, China; 4State Key Laboratory of Powder Metallurgy, Central South University, Changsha 410083, China

**Keywords:** additive manufacturing, characteristic microstructure, steel−based materials, phase transformation, heat treatment

## Abstract

Differing from metal alloys produced by conventional techniques, metallic products prepared by additive manufacturing experience distinct solidification thermal histories and solid−state phase transformation processes, resulting in unique microstructures and superior performance. This review starts with commonly used additive manufacturing techniques in steel−based alloy and then some typical microstructures produced by metal additive manufacturing technologies with different components and processes are summarized, including porosity, dislocation cells, dendrite structures, residual stress, element segregation, etc. The characteristic microstructures may exert a significant influence on the properties of additively manufactured products, and thus it is important to tune the components and additive manufacturing process parameters to achieve the desired microstructures. Finally, the future development and prospects of additive manufacturing technology in steel are discussed.

## 1. Introduction

In recent decades, metal additive manufacturing techniques have received more and more attention [1,2]. The material products fabricated by metal additive manufacturing have been extended, but are not limited, to steel [3,4], aluminum alloy [5,6], magnesium alloy [7,8], titanium alloy [9], high−entropy alloy [10,11], etc. As one of the most fundamental metal materials, extensive research on additive manufacturing of iron−based alloys has been reported in the literature, mainly including 18Ni300 mold steel, 316 stainless steel, 304 stainless steel, etc. Compared with traditional manufacturing techniques (TMT), additive manufacturing methods can effectively save processing time and improve material utilization [12]. Due to the distinguished advantages of additive manufacturing (AM), the technique has been widely applied in industries [13,14,15]. 

The basic principle of AM is similar to that of multi−pass welding technology in that the material powder melts once the heat source passes through and solidifies before the next heat source [16]. In additive manufacturing, the printing parameters exert a significant effect on the characteristic microstructure. For instance, laser−generated melt pools and thermal gradients during solidification may produce columnar grains and textures in the products. In addition, the large temperature gradients cause a non−equilibrium microstructure, which is distinct from the microstructure produced by TMT. After cooling, the secondary heat cycle caused by the newly melted layer covering the solidified layer may further facilitate component diffusion and microstructure evolution [17,18]. The deposition path is another important influencing factor on the performance of AM samples [19,20]. To study the effect of deposition paths on sample performance and productivity, Veiga et al. [21] conducted several strategies. It was found that waving and cross−waving were the most beneficial strategies in terms of productivity, which achieved a 50% torch utilization rate compared to the total time.

The process parameters of additive manufacturing, such as scanning rate, powder particle size, pre−heat temperature, etc., significantly affect the performance of the additively manufactured products [22,23]. To investigate the effect of process parameters on the microstructure evolution and tensile performance of 304L stainless steel by additive manufacturing, Wang et al. [24] prepared two samples with different heat inputs by additive manufacturing. It was found that the samples with low linear heat input displayed higher strength and elongation than the samples with high linear heat input. Helmer et al. [25] also found that the transition of columnar grains to equiaxed grains could be achieved by rapidly switching the orientation of the additive manufacturing heat source and keeping the trajectories of the melted regions overlapped. To investigate the corrosion performance of 316L stainless steel during selective laser melting (SLM) additive manufacturing, Zhao et al. [26] employed SLM equipment to manufacture 316L stainless steel with distinct scanning tactics and then soaked the samples in NaCl aqueous solution (3.5 wt%) for electrochemical tests. It was found that although the passivation film on the side of the 316L stainless steel sample was thicker than the head of the surface, there was more pitting and faster corrosion rates occurred on the sides with more molten pool boundaries. To explore a way to reduce the cost of fine powder in additive manufacturing, Yang et al. [27] mixed the 316L fine and coarse powders with different mass ratios via ball milling, finding that the mechanical properties of the fabricated SLM samples with a mass ratio of 80:20 were comparable to those of the SLM samples fabricated with pure fine powder, which was closely related to the complex coupling of temperature gradients and surface tension gradients during additive manufacturing.

Based on the information retrieved with the keywords “additive manufacturing” and “steel” in the Web of Science database, as shown in Figure 1a, more than 9500 relevant articles on the topic of AM with steel materials have been published in the past 22 years, and the citation frequency of relevant additive manufacturing literature is increasing year by year. In the present work, we introduce the rapid development and advantages of AM and the utilization of additive manufacturing in steels, and then some commonly used compositions and processes of additive manufacturing are summarized. Furthermore, as shown in Figure 1b, steel−based alloys printed by additive manufacturing display some common characteristic microstructures, such as porosity, dendrites, residual stress, composition segregation, etc. These microstructures can be tuned by composition and process parameters, which in turn affect the service performance of the products prepared by additive manufacturing. Finally, the future development prospects and directions of AM in steel materials are discussed.

## 2. Advanced Additive Manufacturing Techniques

AM is a bottom−up, layer−by−layer manufacturing technique based on 3D model data, and the fabricated samples can be constructed by melting powder, wire, etc. using a laser or electron beam [24]. Additive manufacturing techniques with optimized process parameters can produce many superior properties, such as increased strength, improved ductility, and effectively enhanced service performance [28]. These technical methods can produce geometrically complex samples that are difficult to produce by TMT, which is advantageous in integrating digital design and product production. However, it is worth mentioning that high cost is one of the factors limiting the development of AM. The vision of additive manufacturing is to use these techniques to produce complex metal products for critical functions such as turbine blades and jet engines in aerospace [29]. In recent years, the development of AM techniques has been very rapid [30], including DED, powder bed fusion (PBF), binder jetting [31], metal extrusion [32], atomic diffusion additive manufacturing (ADAM) [33], sheet lamination [34,35], material jetting [36], arc additive manufacturing, [37] etc. For steel materials, PBF and DED additive manufacturing techniques are the most extensively used.

The printing parameter settings of AM influence the melting and solidification process of the powder and then determine the microstructures of the AM sample [38]. For instance, excessively high melt pool temperature and the presence of thermal gradients is the main reason for the formation of columnar grains and textures in the samples. Traditionally, casting and forging produce near−equilibrium microstructures, but excessive cooling rates during solidification in additive manufacturing produce non−equilibrium microstructures. Thermal cycling induced by the duplicate deposition of new molten layers on the solidified layer also leads to iterative microstructural evolution [39]. Some key parameters affect specimen quality, including building layer thickness, laser power, hatch spacing, scanning speed, etc. For the additive manufacturing of steel specimens, some commonly used process parameters are listed in Table 1.

### 2.1. Powder Bed Fusion

In additive manufacturing, powder bed fusion is one of the most prevalent techniques that selectively melts/sinters areas of powder beds using thermal energy. Its heat sources mainly include lasers and electron beams. According to these two heat sources, PBF can be separated into two main techniques: SLM using high−intensity lasers and electron beam melting (EBM) using electron beams. In both processes, the powder needs to be held on a build platform [56].

#### 2.1.1. Selective Laser Melting

SLM is a powder bed AM that applies a high−energy laser beam to selectively melt successive layers of powder in order to fabricate a sample [57,58,59]. During irradiation with laser beam energy, the irradiated powder melts and forms a tiny molten pool [60,61]. The thermal history experienced by samples manufactured by SLM technology is different from that of samples that have undergone traditional technologies [62,63]. In contrast to TMT, this technique combines fast melting and solidification, circular heating, and reciprocal cooling of the deposited layer to produce a characteristic microstructure that differs from that obtained by TMT [24,64,65]. The SLM manufacturing process displays the characteristics of a high utilization rate of raw material powder, which has great advantages in the production of complex samples [66]. SLM production sample quality is influenced by many factors, for instance, laser scanning speed, powder and shape, energy input, and so on. The SLM production process involves complex physical processes, such as fast melting and solidification [67,68], absorption and transmission of laser energy [69], as well as material flow in the molten pool [70].

#### 2.1.2. Electron Beam Melting

Selective EBM additive manufacturing builds a sample layer−by−layer in a powder bed by selectively melting the powder using a beam of high−energy electrons [71,72]. Unlike SLM, electron beam application requires electrical conductivity and is therefore only suitable for metallic materials. When printing samples, the speed of the electron beam is as high as 10^5^ m/s, that is, the electron beam can jump from one point to another in an instant. Therefore, by taking advantage of this feature, electron beam additive manufacturing can realize innovative heating and melting strategies.

### 2.2. Direct Energy Deposition

Another commonly used AM is DED. Differing from the SLM process, the DED production process employs a metal wire or metal powder flow instead of a powder bed as the raw material injection, and then it melts and deposits the material on a substrate using an electron beam or laser. Laser−engineered net shape (LENS) is a representative technique of DED technology [73]. LENS is an AM that uses a laser beam to feed metal powders of different compositions and properties into a molten pool for melting. The difference between LENS and SLM is the metal powder addition process. LENS technology adopts a synchronous powder feeding process in the molding process, wherein the metal powder is sprayed and heated by the laser beam at the same time. The LENS process, with the advantages of high molding efficiency and high sample density, can also be sprayed on the surface of the sample [74]. Another development of DED technology is the combination with topology optimization technology to design samples with excellent performance, wherein topology optimization is a mathematical method to allocate materials within a given design range according to specific physical problems and optimization targets [75].

## 3. Characteristic Microstructures of Steel Prepared via AM

Due to the complex phase structure in steel materials, its structure and performance are usually related to the solidification process and thermal history. Consequently, compared with other metal materials, the subsequent heat treatment strongly affects the microstructure and performance of the 3D−printed steel material, and thus further process optimization is very important [76]. Table 2 lists some commonly printed steels in additive manufacturing, including stainless steel, tool steel, die steel, etc. The characteristics of additively manufactured samples are summarized from the aspects of composition, process, and performance.

### 3.1. Porosity in Additively Manufactured Steel

Porosity is an important concern in metal additive manufacturing, as it worsens the apparent strength of the sample and decreases the fatigue life of the products. Among additively manufactured specimens, porosity formation is a hot research topic [106]. The main causes of porosity in additively manufactured specimens are residual gases from metal powder raw materials and powder solidification without fusion, which exerts a notable influence on the corrosion performance of AM samples [107]. Itzhak et al. [108] conducted a study by putting 316L stainless steel into sulfuric acid. The results showed that the porosity in the specimen was the major determinant of anti−corrosion. To study the occurrence of pitting corrosion of stainless steel, Prieto et al. [109] prepared 316L with a direct metal laser sintering process, finding that the residual porosity and microstructural deformation rendered the sample more susceptible to pitting. Laleh et al. [110] observed that the erosion and corrosion of the 316L sample prepared by SLM were poor and it was closely related to the porosity in the sample. To study the sliding wear behavior of 316L, Sun et al. [111] prepared 316L by LSM. It was found that the presence of porosity in the specimen had a more important effect on the wear behavior than the microhardness.

To examine the influence of heat treatment temperatures on the mechanical performance and wear behavior of AM samples, Emre et al. [112] conducted experiments with laser selective melting using three heat treatment temperature methods: HT−1, HT−2, and HT−3 were respectively kept at 600, 850, and 1100 °C for 2 h, respectively, and then air−cooled. Figure 2a presents a schematic drawing of the wear test. The porosity of the four heat−treated specimens is presented in Figure 2b. The porosity of the as−prepared specimen, HT−1, HT−2, and HT−3 was measured by optical microscopy, with values of 0.43%, 0.38%, 0.29%, and 0.08%, respectively. As the heat treatment temperature increased, the sample structure was homogenized and the porosity was further reduced, which was closely related to factors such as process parameter optimization, gas overflow, etc. Figure 2c shows the wear curves of the sample wear test. The figure displays that the various heat treatments exerted a distinct influence on the wear behavior of the samples. The wear depth of the as−prepared sample was approximately 52 µm, whereas the wear depth of the samples with HT−1, HT−2, and HT−3 treatments were 62, 53, and 45 µm, respectively. This study shed light on the fact that the wear resistance of 316L produced by SLM was significantly affected by the porosity, and the wear resistance increased with decreasing porosity.

### 3.2. Dendrite Structures in Additively Manufactured Steel

During the process of printing metal samples via additive manufacturing, dendrite structures are developed. To study the dendritic microstructure and its effect on the mechanical performance of 316L, Chen et al. [113] produced 316L samples using the gas metal arc additive manufacturing (GMA−AM) technique. It was found that austenite dendrites were aligned vertically to the GMA−AM 316L sheet. To investigate the microstructure and its mechanical performance of 308L stainless steel, Le et al. [49] fabricated thin−walled 308L samples by gas−shielded welding additive manufacturing (GMAW−AM). It was found that columnar dendrites mainly existed in the GMAW−AM thin−walled 308L microstructure, and the columnar dendrites grew towards the deposited direction with an increasing number of layers in the printed samples. In addition to the arc additive manufacturing process, cold metal transfer is another potential technique [37].

To investigate the microstructure and the corresponding mechanical performance of additive manufactured 304L steel, Ji et al. [114] conducted tensile tests and metallographic experiments. It was found that as the number of printing layers increased, a slower cooling rate, thicker dendrites, and much more stable dendrite morphology were observed. Figure 3a illustrates a schematic drawing of the developed experimental setup. As shown in Figure 3b, coarse columnar grains appeared on the surface of the chemically etched samples. To better identify the microstructural features, 4 regions were selected within the sample, which were labeled b2, b3, b4, and b5 from top to bottom, respectively. Figure 3b presents the microstructure of dendritic growth within the whole sample, showing that the dendrites became thicker and more stable as the number of layers increased. From top to bottom, the spacing between primary dendrite arms in Figure 3(b2–b5) is 15.44, 13.59, 8.23, and 4.94 um, respectively, and the spacing between the secondary dendrite arms is 7.14, 8.93, 4.98, and 3.52 um, respectively. A total of six tensile specimens (T1, T2, T3, L1, L2, and L3) were fabricated to identify the mechanical performance of the specimen, and the results of the six tensile tests are displayed in Figure 3c. As displayed in Figure 3c, the strength of the T series specimens was higher than that of the L series specimens but the elongation was much smaller than that of the L series specimens. On the whole, the performance of the L series samples was superior. The high strength of the T series samples and high elongation of the L series samples were due to the presence of dendrites, wherein the T series samples were perpendicular to the dendrite and the L series samples were parallel to the dendrite. During the tensile process of the T series samples, the dislocation movement was hindered by the dendrite boundary, which resulted in the increased strength and decreased elongation. Meanwhile, the tensile test of the L series samples displayed an opposite trend.

### 3.3. Dislocation Cells in Additively Manufactured Steel

Unlike conventional manufacturing processes, cellular structures are usually generated inside the grains of some additively manufactured alloys. Due to the high internal stress inside the sample, a high density of dislocations are formed on the cell wall [76], which are usually closely related to the excellent yield strength of additively manufactured samples [115,116].

To study and examine the cellular boundary of SLM 316L stainless steel, Hong et al. [117] employed different heat treatments to tune the cellular sub−grains of laser−melted 316L. It was found that the dislocation density at cellular boundaries exerted a critical role in interfacial strengthening. As shown in Figure 4(a1), the experimental sample size was 5 mm × 20 mm × 80 mm, which was fabricated by SLM equipment, and the tensile specimen is presented in Figure 4(a3). Four different heat treatment temperatures (500, 900, 950, and 1100 °C) were applied for 1 h and then the AM samples were furnace cooled. Ar gas was filled as a protective gas through the heat treatment operation. According to the temperature of heat treatment, the four samples were designated HT500, HT900, HT950, and HT1100, respectively. The TEM images shown in Figure 4b are the as−prepared specimen and the HT1100 sample. The TEM image shown in Figure 4(b1) displays that the cellular sub−grains include a high density of dislocations. Figure 4(b2) presents many dislocations around the cell boundaries. Thermal shrinkage stress during rapid solidification greatly contributed to as−received high dislocation density [118]. The dislocation density shown in Figure 4(b3) was lower than that shown in Figure 4(b1), which demonstrated that the dislocation density at the unit cell boundary was greatly reduced after the specimen was heat−treated. Figure 4(b4) is an enlarged view of the unit cell boundary in Figure 4(b3), showing that the dislocations were uniformly distributed on the unit cell boundaries. Figure 4c shows a set of tensile curves for the SLM samples with various heat treatments; the yield strength reduced from 578 ± 5 MPa for the as−built sample to 326 ± 5 MPa for the HT1100 sample.

### 3.4. Residual Stress in Additively Manufactured Steel

During the additive manufacturing process, the powder layers are melted and solidified layer−by−layer and the expansion and contraction stress accumulates, forming high residual internal stress, which may be released or redistributed during long−term service, resulting in fatigue cracks, brittle fracture, and stress corrosion failure. For example, austenitic stainless steel for nuclear applications is prone to stress corrosion cracking (SCC) in high−temperature water, which is closely related to residual stress. Furthermore, the main factors affecting SCC are temperature, radiation damage, electrochemical potential, water chemistry, sensitization, etc. [119,120,121].

To investigate SCC growth in high−temperature water with 316L manufactured using the laser powder bed method, Lou et al. [122] evaluated the experimental parameters and their influence, including crack orientation, microstructure, and stress intensity factor. Figure 5a shows the direction of the sample compared to the powder bed, while the X and Y axes are parallel to the powder bed and the Z axis is the orientation in which the specimen was built. Z−X indicated crack growth in the X direction and loading in the Z direction. It was found that stress−relieved 316L stainless steel showed two special cracking characteristics: (1) the cracks grew in the build direction; (2) the hydrogen water chemistry did not affect the cracks in the X−Z direction when forging along the X direction. Figure 5b presents the EBSD plot of the crack formation in the AM 316L stainless steel without cold work. From the crystal boundary diagram and inverse pole diagram, it could be seen that some cracks developed along the high−angle boundary of adjacent grains. High−angle grain boundaries were not the only path for SCC propagation, cracks also propagated along sub−grain structures, such as low−angle grain boundaries and dislocations. According to Figure 5c, the SCC growth rate was dependent on the stress intensity factor (*K*) in AM 316L stainless steel under HIP + SA conditions (annealed and 20% cold work). 

### 3.5. Element Segregation in Additively Manufactured Steel

In additive manufacturing, due to the fast cooling rate, the elements in the solidification zone do not have enough time to fully diffuse, thus element aggregation may occur. Elemental segregation between dendrites of additively manufactured samples plays an important influence on corrosion resistance [123,124]. To investigate the effect of linear heat input (LHI) on the microstructure and corrosion behavior of austenitic stainless steel, Wen et al. [123] prepared austenitic stainless steel using wire−arc additive manufacturing, observing that steel with high LHI was strongly related to the segregation of Cr and Mo atoms in Ni−poor δ−ferrite.

To develop a 316L stainless steel (SS) with high yield strength and ductility, Wang et al. [77] employed two different laser powder bed fusion (L−PBF) techniques (‘Concept’ and ‘Fraunhofer’), which showed strength and ductility beyond traditional 316L SS. Figure 6a mainly shows the microstructure of 316L SS fabricated by L−PBF, which includes grain morphology, grain boundary angle, dislocation density, composition segregation, etc. The detailed EBSD analysis is shown in Figure 6(a3), demonstrating a large number of small−angle grain boundaries inside the grains. It can be seen from Figure 6(a4) that the elements Cr and Mo segregated at the cell wall. Elemental segregation and a large number of low−angle grain boundaries enhanced dislocation pinning and promoted twinning, which in turn affected the strength and ductility of the additively manufactured samples. Figure 6b shows the component analysis of the cell wall and cell interior of the PBF 316L SS manufactured by Fraunhofer machines. It can be seen from the figure that there was very little elemental segregation in the Fraunhofer samples. As shown in Figure 6c, both L−PBF−fabricated 316L samples were stronger than the cast and forged samples in terms of strength and ductility. Figure 6(c2) shows a summary of yield stress versus uniform elongation for various 316L SS. The outstanding strength and ductility of 3D−printed steels exceeded that of conventional 316L SS. The excellent performance depended on many factors, including that the compositional segregation at the grain boundary may have pinned the dislocation motion and, meanwhile, a large number of small−angle grain boundaries would also hinder the movement of dislocations, resulting in increased strength.

### 3.6. Other Structural Characteristics in Additively Manufactured Steel

In addition to the above five characteristic microstructures in additively manufactured products, some other characteristic structures also exert an important effect on the service properties of additively manufactured samples. Due to local melting and non−uniform heating during additive manufacturing, there may be defects such as poor powder fusion that could affect the surface roughness of the printed sample. Melting and directional solidification during printing can lead to periodic cracks, which is one of the main limitations of the wide application of additive manufacturing in metallic materials.

The DED technique provides an opportunity to print graded [125] or layered [126] functionally graded materials by changing the powder supply [125]. For instance, changing the percentage of stainless steel to Inconel during printing can create hardness gradients in the specimen [127,128]. Hofmann et al. [129] designed a path from one alloy to another based on a multi−component phase diagram to evade harmful phase formation between the two components. On this basis, using a multi−hopper LMD system, linear and radial gradient alloys could be designed and manufactured.

To study the microstructure and performance of 316L−Inconel 718 composition gradient stainless steel alloy, Wen et al. [130] used a laser powder bed to prepare composition gradient alloy (CGA) 316L and Inconel 718 alloys. It was found that the mechanical properties of the specimens varied with the composition gradient. Figure 7(a1) presents a schematic diagram of the powder bed. Figure 7(a2) shows the shape of the printed sample with the manufacturing direction and compositional gradient direction (GD). The authors mainly studied the composition changes of five main elements, Cr, Mo, Fe, Nb, and Ni, in 316L/IN718 stainless steel constructed along GD. Figure 7b shows the results of the weight percent of elements measured by EDS. Figure 7c displays a cross−section of the sample showing the XRF compositional patterns of the five main components. The gradual change in element color along the GD direction confirmed the existence of a compositional gradient in the CGA. At different positions along the BD orientation of the sample, the consistent color of the elements indicated that the powder deposition process was stable. Various cross−sectional positions of CGA slices were tested for uniaxial tensile performance in two conditions: as−prepared and heat−treated samples. Figure 7(d1) and Figure 7(d2) show the relationship between engineering stress and engineering strain for IN718 at 0, 8, 25, 48, 65, 82, and 100 wt% cross−sections, respectively. In the slices with 0, 8, and 25 wt% IN718, σ_y_ gradually decreased. The results indicated that σ_y_ was reduced with increased IN718 content at low IN718 content (≤25 wt%). As the proportion of IN718 increased to 48 wt%, σ_y_ did not decrease but increased to 647 ± 28 MPa. Then, σ_y_ and UTS significantly increased with further increase in IN718 content. When the IN718 proportion further increased to ≥82 wt%, YS and UTS were higher while ductility was lower in the CGA sample relative to the as−prepared sample.

## 4. Conclusions and Perspectives

In summary, AM has been widely used in the design and manufacture of high−performance steels, effectively saving processing time and improving material utilization; however, the physical metallurgical process of additively manufactured steel is very complex. This work summarizes some typical microstructures of additively manufactured steel−based alloys. Specifically, the high−temperature gradients in additive manufacturing processes can form dislocation cells and alloying elements may segregate and aggregate at defects; the repeated melting and solidification of the powder layer can cause large residual stress; improper processing parameters of the laser can generate pores in the specimen, etc. These characteristic microstructures exert a significant influence on the properties of additively manufactured products. The characteristic microstructures summarized in this work will be helpful for follow−up research, and this work may promote the application of additive manufacturing technology in the field of steel−based alloys. In recent years, the development of AM has presented a diversified scene, considering the multi−scale and complex phase transformation characteristics of the steel itself. Nevertheless, there are some trends in developing high−performance steel−based materials via additive manufacturing.

The additive manufacturing technique is a non−equilibrium solidification process, and the microstructure structure exhibits multi−level and cross−scale characteristics. It is difficult to quantitatively characterize the microscopic mechanism of additively manufactured products in experiments. Therefore, it is highly desirable to develop advanced multi−scale computing techniques to shed light on the complex mechanism of microstructure evolution and thus improve the macro−performance.As a potential high−throughput experimental method, the additive manufacturing technique can effectively accelerate the composition and process optimization design of high−performance steel−based materials by gradient printing.Steel is born with complex solid−state phase transition; therefore, learning from the abundant traditional heat treatment experience and developing a heat treatment scheme suitable for additive manufacturing is one of the future research directions.At present, all grades of steel are proposed for traditional steel preparation processes, but it is urgent to establish a set of steel grades suitable for additively manufactured steel.Data−driven additive manufacturing technology is another future direction. Unlike traditional steel preparation, 3D printing of metal specimens lacks a large amount of high−quality data at present, thus it is also urgent to develop a database and data−driven strategies for additive manufacturing.

## Figures and Tables

**Figure 1 materials-16-02696-f001:**
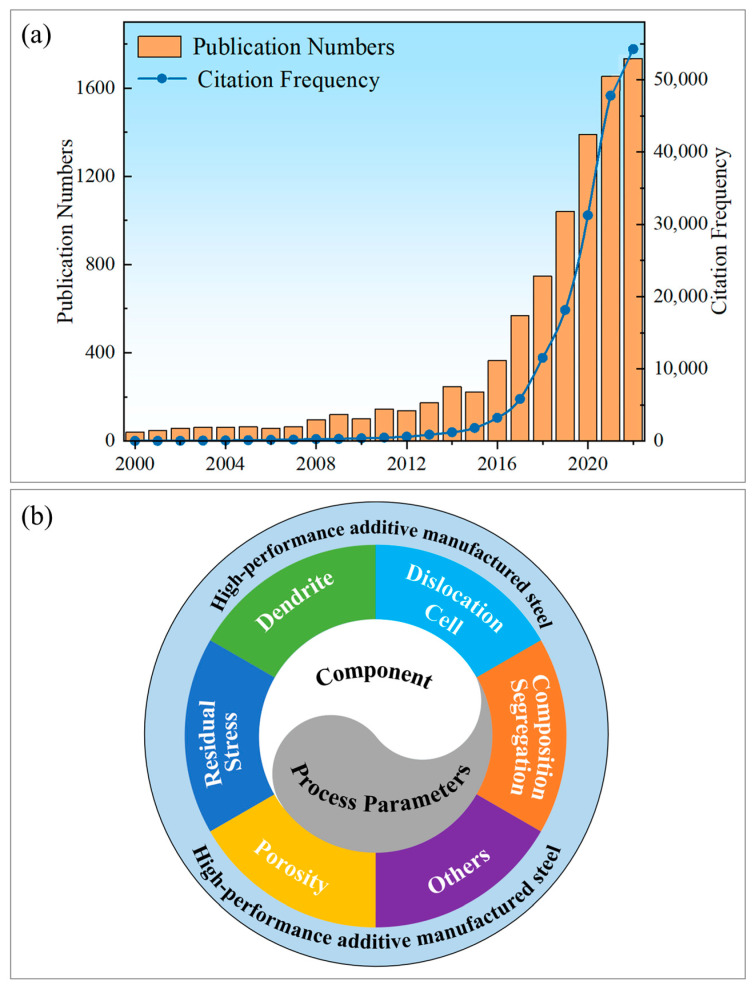
Utilization of additive manufacturing in steel−based alloys. (**a**) Histogram of the number and citation frequency of relevant articles retrieved with the keywords “additive manufacturing” and “steel” in the Web of Science database. (**b**) Some typical microstructures produced by additive manufacturing in steels, and the relationship among the component, process, microstructure and desired performance.

**Figure 2 materials-16-02696-f002:**
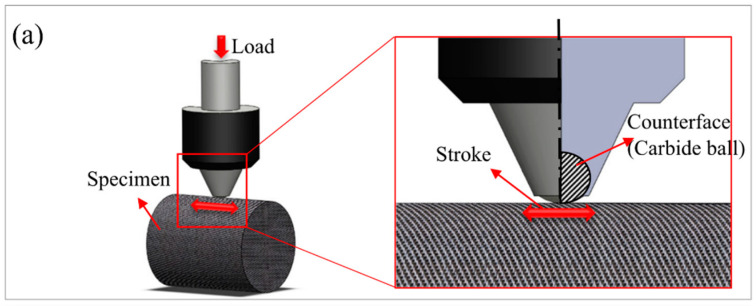
The influence of heat treatment temperature on the porosity of additively manufactured 316L. (**a**) Schematic drawing of the wear test; (**b**) porosity in different samples: (**b1**) as−built sample, (**b2**) HT−1 sample, (**b3**) HT−2 sample, (**b4**) HT−3 sample; (**c**) wear profiles comparing stainless steel 316L of as−built, HT−1, HT−2, and HT−3 treatments. (Reprinted with permission from ref. [112]. Copyright 2020, International Journal of Advanced Manufacturing Technology).

**Figure 3 materials-16-02696-f003:**
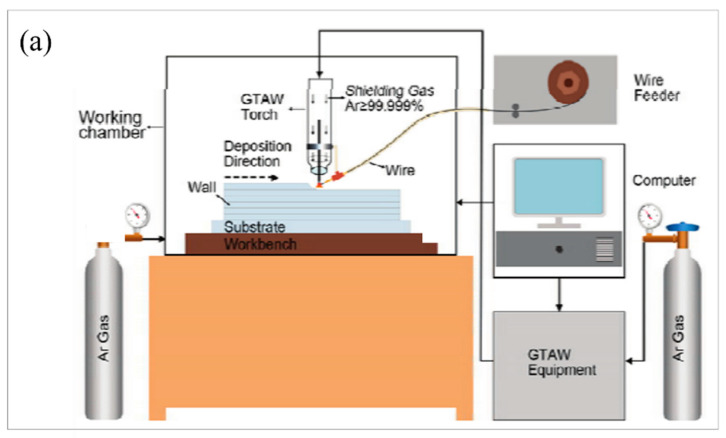
The influence of dendritic structure on the mechanical properties of the additively manufactured 304L stainless steel products. (**a**) Schematic drawing of the experimental device; (**b**) sample structure diagram (**b1**) and metallographic micrographs of four regions (**b2**–**b5**); (**c**) stress−strain curves of the longitudinal (L1, L2, L3) and transverse (T1, T2, T3) tensile specimens. (Reprinted with permission from ref. [114]. Copyright 2017, Electronic Material).

**Figure 4 materials-16-02696-f004:**
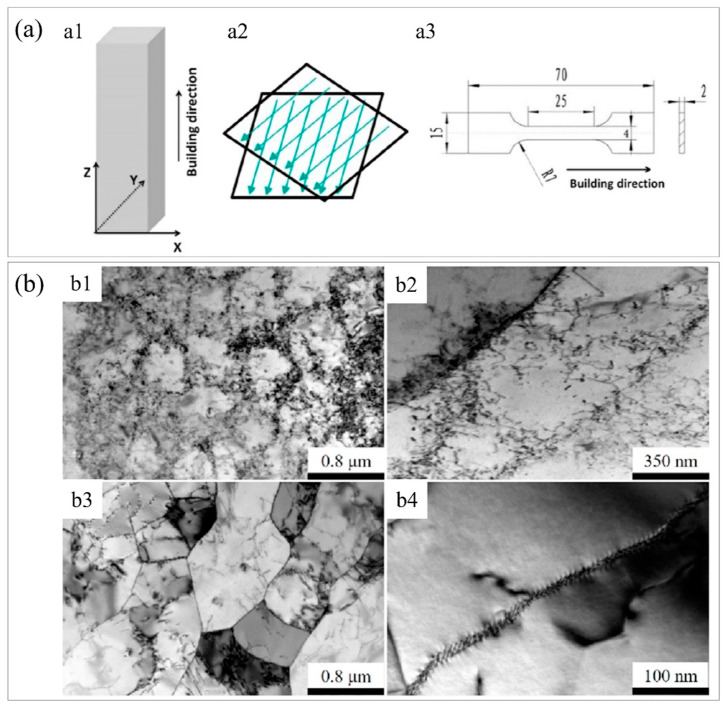
The effect of dislocation density on the performance of additively manufactured 316L. (**a**) Experimental results and process parameters, (**a1**) schematic diagram of the sample fabricated by SLM, the X and Y axes denote directions equal to the powder bed, whereas the Z axis represents the sample build direction, (**a2**) laser scanning strategy of SLM, (**a3**) schematic diagram of the tensile sample size. (**b**) TEM images of as−prepared and HT1100 samples, (**b1**) TEM image of as−prepared sample, (**b2**) TEM image of an enlarged view of (**b1**), (**b3**) TEM image of HT1100 sample, (**b4**) TEM image of an enlarged view of (**b3**). (**c**) Tensile stress−strain curves for the studied samples. (Reprinted with permission from ref. [117]. Copyright 2021, Materials Science Engineering: A).

**Figure 5 materials-16-02696-f005:**
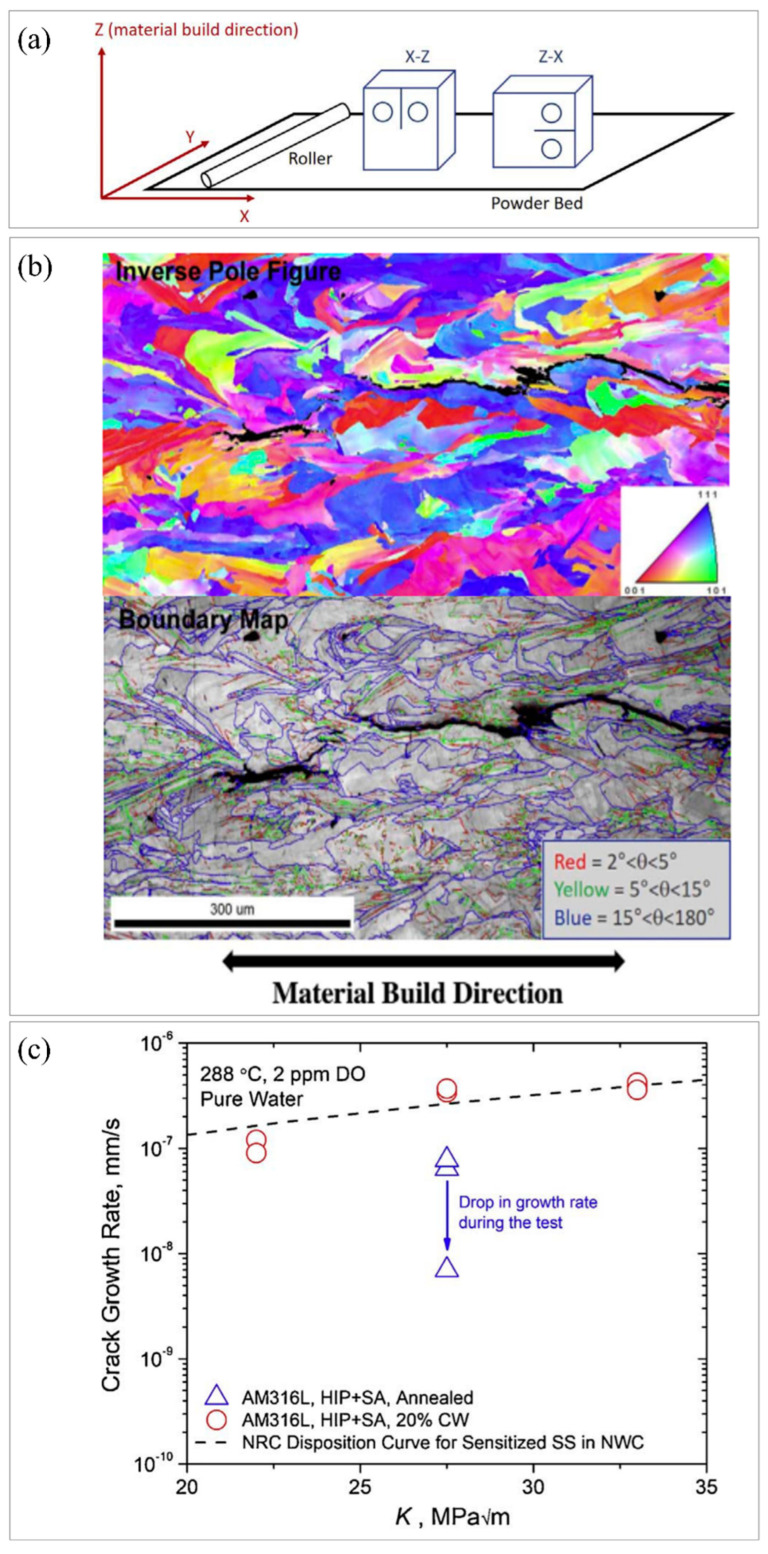
In high−temperature water, SCC develops in additively manufactured stainless steel. (**a**) Schematic drawing of the direction of the tensile specimen compared to the powder bed, the X and Y axes denote the directions parallel to the powder bed, while the Z axis represents the sample build direction. (**b**) EBSD plot of SCC in stress−relieved additively manufactured 316L without additional cold working in the X−Z direction, including inverse pole figure and grain boundary map. (**c**) The influence of stress intensity *K* on the growth rate of SCC in AM 316L stainless−steel in normal water chemistry. (Reprinted with permission from ref. [122]. Copyright 2017, Corrosion Science).

**Figure 6 materials-16-02696-f006:**
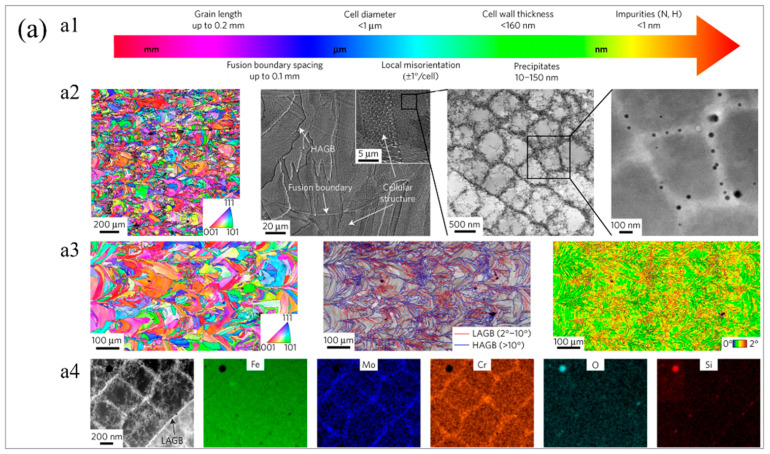
Typical microstructure and mechanical properties of 316L SS prepared by L−PBF. (**a**) Typical microstructures of 316L SS produced by L−PBF, (**a1**) illustration of the multi−scale microstructures in 316L SS, (**a2**) cross−sectional EBSD map, SEM image, bright−field TEM image, and dark−field STEM image of t316L SS, (**a3**) detailed EBSD analysis, (**a4**) HAADF STEM (Z contrast) image with corresponding Cr Fe, and Mo EDS maps showing Cr and Mo segregation at the solidification cellular walls. (**b**) Compositional analysis of cell walls and cell interior for L−PBF 316L SS fabricated using the Fraunhofer machine, (**b1**) bright−field TEM image of cells, (**b2**) high−angle annular dark field image with corresponding elemental maps, (**b3**) detailed composition of the selected regions. (**c**) Mechanical properties of L−PBF 316L SS, (**c1**) tensile engineering stress−strain curves, (**c2**) summary of yield stress vs. uniform elongation for various 316L SS, wherein the references in this figure should be referred to the original article [77]. (Reprinted with permission from ref. [77]. Copyright 2018, Nature Materials).

**Figure 7 materials-16-02696-f007:**
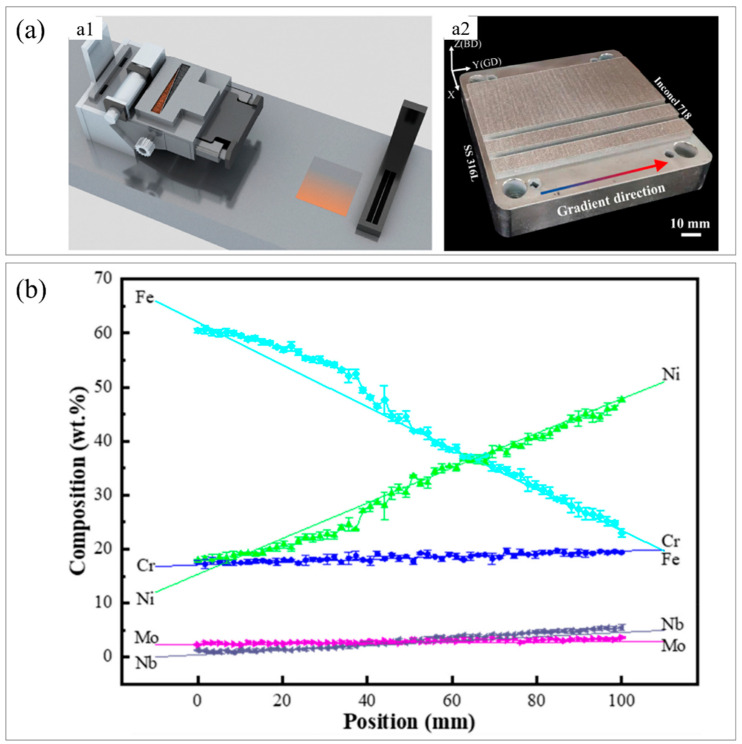
Composition variation of gradient specimens prepared by powder bed. (**a**) Schematic diagram and sample image, (**a1**) schematic illustration of the CGA LPBF system, (**a2**) image of fabricated CGA samples; (**b**) component distribution map measured by EDS technology along the gradient orientation; (**c**) composition gradient distribution map of CGA measured by XRF on the plane; (**d**) engineering stress−strain curves of (**d1**) as−prepared and (**d2**) heat−treated CGA specimens. (Reprinted with permission from ref. [130]. Copyright 2022, Materials Science Engineering: A).

**Table 1 materials-16-02696-t001:** List of several typical deposition techniques and the relevant parameters in AM steel.

Process	Material Shape	Travel Speed (mm s^−1^)	Spot Size (mm)	Layer Height (mm)	Heat Input (W)	Material Feed Speed (mm s^−1^)	Ref.
Laser DED	powder	2.5–20	1.2–2	0.25–0.5	360–2600	2–20.4	[40,41,42,43,44,45]
PTA		1.3–1.7	/	/	/	25–35	[46]
GMAW	wire	2.5–30	/	0.5–2	3500–8400	28–166	[47,48,49,50,51,52,53,54]
GTAW		2.92–7	/	/	1920	16.67–58	[55]
PTA		0.6–2	/	/	350–3510	9–28	[55]

**Table 2 materials-16-02696-t002:** List of composition, process, and properties of metal materials printed by AM. (SS: stainless steel, AP: as−produced, SA: solution annealed, AH: aging heat−treated, BP: base plate temperature during the preparation process.).

Steel Type	Elements (wt%)									3D Printing Techniques	Heat Treatment Process	Mechanical Properties				Ref.
	C	Cr	Ni	Mo	Mn	Si	Ti	Al	Others			YS (MPa)	UTS (MPa)	Elongation (%)	Hardness (HV or HRC)	
316L	<0.03	16–18	10–14	2–3	<2	<0.75	/	/	N < 0.1	L−PBF	AP	450	640	59		[77]
										L−PBF	AP	590	700	36		
										L−DED	AP	470	675	52.5		[78]
										L−DED	AP	535	665	35		
										L−DED	AP	405	655	57		
										L−DED	AP	505	670	41.58		
										L−PBF	AP	602	664	30		[79]
										L−PBF	AP	557	591	42		
										L−PBF	AP	534	653	16.2		[80]
										L−PBF	AP	444	567	8		
										L−DED	AP	490	685	51		[42]
										L−DED	AP	280	580	62		
17-4 PH	<0.07	15–17.5	3–5		<1.0	<1.0	/	/	Nb 0.15–0.45	L−PBF	L−PBF	452	1119	15.2		[81]
										L−PBF	AP	798	1101	15.8	346.3 HV	[82]
										L−PBF	AP	824	916	4.2	356.1 HV	
										L−PBF	AP	810	948	4.8	350.2 HV	
										L−PBF	AP	773	1043	17.6	355.3 HV	
										L−PBF	AP	873	951	5.3	346.7 HV	
										L−PBF	AP	866	935	3.3	350.3 HV	
										L−PBF	AP	1190	1370	8.3	380 HV	[83]
										L−PBF	AP	570	944	50		[84]
18Ni−300	<0.03	<0.5	17–19	4.5–5.2	<0.1	<0.1	0.6–0.8	0.05–0.15	Co 8.5–9.5	L−PBF	AP	815–1080	1010–1205	8.3–12	420 HV	[85]
										L−PBF	SA	800	950	13.5	320 HV	
										L−PBF	AH	1750	1850	5.1	600 HV	
										L−PBF	AP		1085–1192	5–8	33 HRC	[86]
										L−PBF	AP	985	1152	7.6	34 HRC	[87]
										L−PBF	AP	915	1188	6.1	371 HV	[88]
										L−PBF	AH	1957	2017	1.5	600 HV	
										L−PBF	AP		1290	13.3	40 HRC	[89]
										L−PBF	AH		2217	1.6	58 HRC	
										L−PBF	AP	915	1165	12.4	35 HRC	[90]
										L−PBF	AH	1967	2014	3.3	54 HRC	
										L−PBF	SA	962	1025	14.4	28 HRC	
										L−PBF	SA+AH	1882	1943	5.6	53 HRC	
H13	0.32–0.45	4.75–5.5		1.1–1.75	0.2–0.6	0.8–1.2	/	/	V0.8 –1.2	L−PBF	AP	1003	1370	1.7	59 HRC	[91]
										L−PBF	AH	1580	1860	2.2	51 HRC	
										DED	AP	1288–1564	2033–2064	5–6	660 HV	[92]
										L−PBF	AP(BP 240 °C)	892	1440	1.5	575 HV	[93]
										L−PBF	AP	1236	1712	4.1		[94]
										L−PBF	AP(BP 200 °C)	835	1620	4.1		
										L−PBF	AP(BP 400 °C)	1073	1965	3.7		
										L−PBF	AP 100 °C	1150–1275	1550–1650	1.5–2.25		[95]
Ferritic SS 441	<0.03	18	<1.0		<1	<1	/	/	Nb < 0.9, Ti0.1–0.5	L−PBF	AP	679	874	30		[96,97]
										L−PBF	AP	741	896	28		[96,98]
Duplex SS 2205	<0.03	21–23	4.5–6.5	2.5–3.5	<2.0	<1.0	/		N0.08–0.2	L−PBF	AP	950	1071.3	16		[99]
										L−PBF	AP		940	12		[100]
Duplex SS 2507	<0.03	24–26	6–8	3–5	<1.2	<0.8			Cu < 0.5, N 0.24–0.32	L−PBF	AP	1214	1321	8	450 HV	[101,102]
Other steels	Also includes duplex stainless steel (SAF2705), ODS steel (PM200), tool steel (M2), etc.															[103,104,105]

## Data Availability

Not applicable.

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
