# Peer review of "The Characteristic Microstructures and Properties of Steel-Based Alloy via Additive Manufacturing"

_materials, 2023, doi:10.3390/ma16072696_

Round 1

Reviewer 1 Report

The manuscript entitled “materials-2241737” dealing with metal printing has been reviewed. The paper has been nicely written but needs significant improvement. Please follow my comments.

1.     What is the main research question for this research work?

2.     What is the reference for Figure 1? The authors need to provide more detail about it.

3.     What is the future direction of this work?

4.     Please update the introduction with the new publications in the field. Authors are encouraged to read and add the following two new papers in the field.

·       Microstructure simulation and experimental evaluation of the anisotropy of 316 L stainless steel manufactured by laser powder bed fusion

·       Material extrusion additive manufacturing of 17–4 PH stainless steel: effect of process parameters on mechanical properties

5.     More explanation about Figure 3 “Figure 3. The influence of dendritic structure on mechanical.” is needed.

6.     Please proofread the paper.

7.     Additive manufacturing has many advantages over the conventional manufacturing method which can be highlighted in your paper. Please read the following manuscript and add it to the literature to show how additive manufacturing is comparable with conventional manufacturing.

“Laser subtractive and laser powder bed fusion of metals: review of process and production features”

Reviewer 2 Report

Please find my comments attached.

Reviewer 3 Report

The paper “The characteristic microstructures and properties of steel-based alloy via additive manufacturing” a review on the application of additive manufacturing on steel-based alloys and effect on properties. The review is well presented and shows interesting points, so it would be advisable to publish it after changes. Some suggestions:

·        In the abstract, I don't think I would use the term "the high cooling rate" which presupposes a certain cooling action, I think it is more correct to refer to the "high temperature gradients".

·        More work should be done on the effect of metal transfer agents on properties of Additive Manufacturing steel in addition to the amount of shielding gas.

·        Fig.4 text size is too small. Text captions are difficult to read.

·        In the introduction, please introduce the effect of the path on the materials:

o   https://doi.org/10.1016/j.jmapro.2022.10.039

o   https://doi.org/10.1016/j.promfg.2020.05.158

·        Another topic with recent contributions that would be interesting to discuss is that of topological optimisation and the vision given by DED.

·        It should be included or at least mentioned that it is out of the study results in metal extrusion, ADAM (Atomic Diffusion Additive Manufacturing).

·        The conclusions need to be reworked, more concrete.

·        Although a small outline of the perspectives has been given but could be explained in depth.

Round 2

Reviewer 1 Report

The paper is ready to publish.